# Design, Synthesis and Evaluation of Novel Molecular Hybrids between Antiglaucoma Drugs and H_2_S Donors

**DOI:** 10.3390/ijms232213804

**Published:** 2022-11-09

**Authors:** Rosa Sparaco, Valentina Citi, Elisa Magli, Alma Martelli, Eugenia Piragine, Vincenzo Calderone, Giorgia Andreozzi, Elisa Perissutti, Francesco Frecentese, Vincenzo Santagada, Giuseppe Caliendo, Beatrice Severino, Angela Corvino, Ferdinando Fiorino

**Affiliations:** 1Dipartimento di Farmacia, Università degli Studi di Napoli “Federico II”, Via D. Montesano 49, 80131 Napoli, Italy; 2Dipartimento di Farmacia, Università di Pisa, Via Bonanno 6, 56126 Pisa, Italy; 3Dipartimento di Sanità Pubblica, Università Degli Studi di Napoli Federico II, Via Pansini 5, 80131 Napoli, Italy

**Keywords:** glaucoma, hydrogen sulfide, H_2_S donors, antiglaucoma drugs, molecular hybrids

## Abstract

Glaucoma is a group of eye diseases consisting of optic nerve damage with corresponding loss of field vision and blindness. Hydrogen sulfide (H_2_S) is a gaseous neurotransmitter implicated in various pathophysiological processes. It is involved in the pathological mechanism of glaucomatous neuropathy and exerts promising effects in the treatment of this disease. In this work, we designed and synthetized new molecular hybrids between antiglaucoma drugs and H_2_S donors to combine the pharmacological effect of both moieties, providing a heightened therapy. Brinzolamide, betaxolol and brimonidine were linked to different H_2_S donors. The H_2_S-releasing properties of the new compounds were evaluated in a phosphate buffer solution by the amperometric approach, and evaluated in human primary corneal epithelial cells (HCEs) by spectrofluorometric measurements. Experimental data showed that compounds **1c**, **1d** and **3d** were the hybrids with the best properties, characterized by a significant and long-lasting production of the gasotransmitter both in the aqueous solution (in the presence of L-cysteine) and in the intracellular environment. Because, to date, the donation of H_2_S by antiglaucoma H_2_S donor hybrids using non-immortalized corneal cells has never been reported, these results pave the way to further investigation of the potential efficacy of the newly synthesized compounds.

## 1. Introduction

The glaucomas are a group of eye diseases characterized by damage of the optic nerve with corresponding loss of field vision [1]. Glaucoma is the leading cause of irreversible blindness and, to date, 11 million people went blind because of this disease [2]. With advancing age, the likelihood of developing glaucoma is higher; therefore, due to the rapid increase in aging population, by 2040 the number of individuals with glaucoma is projected to grow up to 111.8 million [3].

Although the main cause of this neuropathy is unknown and the pathogenesis is not completely understood, intraocular pressure (IOP) is the major modifiable risk factor and is regulated by the balance of aqueous humor (AH) production and outflow. In a healthy human eye, under steady-state conditions, IOP ranges from 10 to 21 mmHg [4]. Usually, in patients with glaucoma, there is an increase in IOP due to a reduced outflow facility of aqueous humor [5]. 

Glaucomas can be classified into open-angle glaucoma (OAG) and angle-closure glaucoma (ACG) [1,5,6,7,8,9]. In eyes with open-angle glaucoma, there are no clinically visible perturbations in the eye and the aqueous humor is free to leave the globe; in contrast, with angle-closure glaucoma, the AH drainage is anatomically reduced or blocked [6]. 

In addition to elevated intraocular pressure, there are common risk factors for the development of glaucomas, such as age, ethnicity, family history of glaucoma, systemic hypertension and diabetes mellitus [5,10,11]. 

Treatment of glaucoma neuropathies aims to reduce IOP, and on the basis of causes, risk factors, severity and type of glaucoma, different medical options such as topical therapy, oral therapy, surgery or laser procedure are available. First-line treatment consists of the topical application through eye drops of IOP-lowering drugs in monotherapy or as drug combinations. Different classes of medications are used to treat glaucoma, and they either increase the outflow of AH from the eye (prostaglandin analogs and cholinomimetics) or reduce its formation (α_2_-adrenergic agonists, β-adrenergic antagonists and carbonic anhydrase inhibitors) [5,7]. 

H_2_S is a colorless, flammable and pungent gas and it has been recognized as a third endogenous gasotransmitter, besides nitric oxide (NO) and carbon monoxide (CO) [12]. Several studies have shown that it plays a role in different physiopathological processes; for example, it acts as cardioprotective agent [13], modulates inflammation [14], reduces oxidative stress [15], induces bronchial relaxation [16] and provides a cytoprotective effect [17].

Despite the interesting properties of H_2_S in the human body, as a gaseous compound, it cannot be considered an ideal drug. For this reason, scientists worked on the development of molecules able to release endogenously H_2_S (named H_2_S donors), that could be used as biological instruments and potential drugs [13]. The most promising H_2_S-releasing compounds are the synthetic donors, characterized by an enhanced safety profile and a better pharmacokinetic profile that mimic the time course of the physiologic H_2_S release. 

The discovery of the enzymes that mediate H_2_S production in ocular tissues suggested a potential physiological role for this gasotransmitter in the eye [18]. Different ocular diseases related to retinal degeneration like glaucoma, AMD (age-related macular degeneration) and DR (diabetic retinopathy) are characterized by the reduction of endogenous H_2_S levels and expression of H_2_S synthetizing enzymes [19]. Several studies have shown that exogenous H_2_S released by molecular donors can reduce RGCs’ damage related to oxidative stress and elevated hydrostatic pressure [15,20,21,22,23]. The vasorelaxant effect associated with H_2_S has also been widely demonstrated in ocular vasculature, improving blood flow in the eye [15,21]. In addition, H_2_S plays a role in ocular structures implicated in AH production and outflow as well as in IOP control [18].

In the last decades, H_2_S-releasing molecules have been linked to several pharmaceutical active compounds to synthetize novel molecular hybrids with the purpose of associating the functionality of the parent drugs and endogenous H_2_S. An interesting example of an antiglaucoma drug conjugated to a H_2_S donor is ACS67, a molecular hybrid of latanoprost acid and ADT-OH, a derivative of anethole dithiolethione. Studies confirmed the potentiality of this drug that combines the IOP-lowering effect of latanoprost and the neuroprotective activity of H_2_S, released by ADT-OH [24]. 

On the basis of the data reported above, and considering the expertise of our research group in the field of H_2_S donors and their applications to synthetize novel chemical entities [25,26,27,28], in this experimental work we designed, synthetized and characterized new molecular hybrids between drugs for the treatment of glaucoma and H_2_S-donating moieties. The aim was to synthesize a compound which combines the action of antiglaucomatous drugs and H_2_S released by donors. The idea was to enhance the efficacy of the IOP-lowering medications with the promising effect of H_2_S to provide a heightened therapy. The molecular hybrids must be stable enough to be administered, but once absorbed in the eye they undergo in vivo metabolic reactions that trigger the disintegration of the hybrids, allowing the antiglaucoma drug and the H_2_S donor to interact with their biological targets. By the application of these new entities, we expect a reduction in the administered dosage and side effects. 

For the synthesis of the new molecular hybrids, amongst the different classes of antiglaucomatous drugs, brinzolamide ((4R)-4-(ethylamino)-2-(3-methoxypropyl)- 1,1-dioxo-3,4-dihydrothieno[3,2-e]thiazine-6-sulfonamide), betaxolol (1-[4-[2-(cyclopropylmethoxy)ethyl]phenoxy]-3-(propan-2-ylamino)propan-2-ol) and brimonidine (5-bromo-N-(4,5-dihydro-1H-imidazol-2-yl)quinoxalin-6-amine) were selected as native compounds for their higher reactivity and affordable cost. These parent agents were linked to the effective H_2_S-releasing molecules already described, such as 4-hydroxybenzothioamide (TBZ) [29], 5-(4-hydroxyphenyl)-3H-1,2-dithiole-3-thione (ADT-OH) [24,25,28], S-ethyl 4-hydroxybenzodithioate (HBTA) [27] and 4-hydroxyphenyl isothiocyanate (HPI) [30]. In addition, an acetic or succinic spacer was introduced as a linker between the two parts to facilitate the formation of the hybrid. 

## 2. Chemistry

Chemical structures of compounds **1a**–**1d**, **2a**–**2d** and **3a**–**3d** are represented in Table 1. The synthetic routes for the synthesis of molecular hybrids of brinzolamide (**1a**–**1d**), betaxolol (**2a**–**2d**) and brimonidine (**3a**–**3d**) are summarized, respectively, in Figure 1, Figure 2 and Figure 3.

The general procedure for the synthesis of compounds **1a**–**1c** and **2a**–**2c** is as follows: brinzolamide **1** or betaxolol **2** solubilized in DMF were condensed with the H_2_S donors previously conjugated to an acetic spacer (compounds **7b**–**9b**), by TBTU coupling in the presence of HOBt and N,N-diisopropylethylamine. 

The synthesis of compound **1d** started from the conversion of brinzolamide in its succinic derivative **4** by treatment with succinic anhydride in acetonitrile. The following coupling reaction of intermediate **4**, solubilized in DMF, with HPI **10** was performed using TBTU, HOBt and N,N-diisopropylethylamine as coupling agents. 

Unlike compounds **2a**–**2c**, for compound **2d**, the spacer-H_2_S donor moiety is linked to the alcoholic function of betaxolol by an ester bond, instead of to the aminic group, due to the unsuccessful reaction between HPI and tert-butyl bromoacetate. In this case, betaxolol **2** was treated with an excess of succinic anhydride and a catalytic amount of DMAP in anhydrous CH_2_Cl_2_ to produce the corresponding hemisuccinated ester that was linked to HPI **10** in the presence of EDAC·HCl and DMAP, obtaining the compound **2d.**

Via one-pot reaction, brimonidine **3** solubilized in anhydrous DMF was first converted into its derivative **5** by treatment with succinic anhydride and DMAP, and then the obtained intermediate was linked to the H_2_S donors (**7**–**10**) by means of EDAC·HCl and DMAP, producing the corresponding compounds **3a**–**3d**. 

The H_2_S-releasing moieties **7** and **10** were commercially available. ADT-OH **8** was synthetized by reacting trans-anethole and sulfur in DMF according to a process reported in the literature [31]. The H_2_S donor HBTA **9** was obtained following the synthetic procedure described by our research group [27]. 

Figure 4 reports the synthetic route for introducing an acetic spacer on the H_2_S donors **7**–**9**. TBZ, ADT-OH and HBTA were reacted with tert-butyl bromoacetate in the presence of NaH in DMF to produce intermediates **7a**–**9a**, which were successfully deprotected by treatment with a 10% (*v*/*v*) TFA solution in CH_2_Cl_2_ affording the desired intermediates **7b**–**9b**. 

## 3. Results and Discussion

### 3.1. Amperometric Evaluation of H_2_S Release

The H_2_S-generating properties of the compounds **1a**–**1d**, **2a**–**2d** and **3a**–**3d** were evaluated by amperometry, allowing a “real time” detection of the released H_2_S with high sensitivity and selectivity [32]. The assay was performed in an aqueous phosphate buffer, in the absence or in the presence of L-cysteine, whose thiol group mimics the endogenous free thiols in the cells. In Table 1, the C_max_ values are reported, representing the highest concentration of H_2_S (μM) recorded during the experiments and released by the H_2_S donating moieties and molecular hybrids (100 μM) in the experimental conditions. 

As illustrated in Figure 1, the amperometric assay demonstrated that in the absence of L-cysteine, all the compounds had a completely negligible release of H_2_S (<0,4 μM), except ADT-OH **8** and HPI **10**. These data proved that the presence of a thiol group activates and/or enhances the H_2_S generation from the tested compounds. Therefore, they act as “smart H_2_S donors” since these agents can donate the gaseous transmitter only in a biological environment, i.e., in the presence of organic thiols [26,32,33]. Otherwise, ADT-OH **8** and HPI **10** were able to release H_2_S both in the absence and in the presence of L-cysteine, due to their susceptibility to both a hydrolytic and thiol-dependent mechanism of release.

All brinzolamide hybrids (compounds **1a**–**1d**) showed an L-cysteine-dependent generation of H_2_S. Nevertheless, the hybrid brinzolamide–TBZ (**1a**) had the lowest release (C_max_ = 0.4 ± 0.2 μM) while the compound **1c** (brinzolamide–HBTA) showed a slow and considerable production of H_2_S and within the series of the brinzolamide hybrids, demonstrated the highest C_max_ (3.5 ± 0.3 μM). Amperometric data obtained from **1c** confirmed the promising results collected by our research group [27], suggesting HBTA as an innovative and effective thiol-triggered H_2_S donor. 

The molecular hybrids synthetized, starting from betaxolol (**2a**–**2d**), had a weak H_2_S release, enhanced by the presence of L-cysteine. The curves for H_2_S release vs. time in the absence or in the presence of L-Cys for compound **2a** (betaxolol-TBZ) were almost overlapping.

Compounds **3a**–**3d** required the presence of L-Cys to obtain a detectable generation of H_2_S. The hybrid brimonidine-HPI **3d** showed the best releasing profile, with progressive and time-related slow gas donation. The compound **3d** produced a significant H_2_S generation with a C_max_ value of 6.2 ± 0.5 μM. In addition, in this case, data from amperometric assay corroborated the studies indicating that isothiocyanates are promising H_2_S donors [32,33]. Furthermore, as illustrated by Lin et al., the endogenous H_2_S release from isothiocyanates occurs in the presence of thiols (mainly GSH or L-Cys). In particular, the authors showed that isothiocyanates react rapidly with the L-Cys to form an adduct, which then undergoes an intramolecular cyclization reaction to finally release H_2_S [34]. In addition, the electronic effect of the substituents linked to the isothiocyanate may influence the H_2_S formation rate. 

### 3.2. Intracellular H_2_S Release in HCEs 

The H_2_S-releasing properties of the novel molecular hybrids were also tested in human primary corneal epithelial cells (HCEs) to verify the H_2_S formation into the cellular environment without adding any exogenous thiol. This method allows one to understand the behavior of the H_2_S donors in the presence of a physiological level of intracellular L-cysteine, since we use non-immortalized corneal cells.

The detection of intracellular H_2_S was performed by spectrofluorometric measurements using the dye 3′-methoxy-3-oxo-3H-spiro-(isobenzofuran-1,9′-xanthen)-6′-yl-(pyridin-2-yldisulfanyl) benzoate (Washington State Probe-1, WSP-1). WSP-1 can react specifically and irreversibly with H_2_S generated by the tested compounds, releasing a fluorophore detectable with a spectrofluorometer. The increase in the fluorescence compared to the blank is expressed as fluorescence index (FI) [33]. The FI values of the H_2_S donors and the hybrids were compared to the FI value of diallyl disulfide (DADS), considered as reference sulfide donor and responsible for significant H_2_S production (*p* < 0.001). The addition of the vehicle (1% DMSO) in the experimental conditions reflects the endogenous production of H_2_S in the cells. 

The experiments were performed in HCEs because the cornea is the major route for topical ocular drug absorption and the corneal epithelium is the most anterior layer of the cornea as well as the main barrier for drug absorption from the tear fluid to the anterior chamber of the eye [35,36]. 

In Figure 2, H_2_S formation values of the donating moieties (100 μM) **7**–**10** are represented. The compounds TBZ **7** and ADT–OH **8** incubated in HCEs led to a weak and not significant H_2_S-release, almost comparable to that of the vehicle, showing their inability to enter the cell and produce H_2_S. On the other hand, HBTA **9** and HPI **10** promoted an elevated and significant (*p* < 0.001) increase of WSP-1 fluorescence, comparable to the reference H_2_S donor DADS. The graphs reporting the histograms of the intracellular H_2_S release after the incubation of the compounds are subjected to area-under-the-curve analysis of the fluorescence increase monitored for 50 min (For a better characterization and for a better comprehension of the results, see the graphs of the kinetic included in the Appendix A).

The intracellular H_2_S-releasing profiles of compounds **1a**–**1d** (100 μM), reported in Figure 3, show that all brinzolamide hybrids evocated a significant H_2_S release. Interestingly, compound **1a** (brinzolamide–TBZ) had an enhanced H_2_S production compared to the TBZ-free moiety. The addition of the hybrid brinzolamide–ADTOH (**1b**) in HCEs caused a higher increase in FI value than ADT–OH by itself. The incubation of compounds **1c** and **1d** (brinzolamide–HBTA and brinzolamide–HPI, respectively) promoted a significant intracellular H_2_S release (*p* < 0.001). 

Compounds **2a** betaxolol-TBZ and **2b** betaxolol-ADTOH did not cause any significant increase in fluorescence (Figure 4). The addition of molecular hybrids betaxolol–HBTA (**2c**) and betaxolol–HPI (**2d**) to WSP-1-preloaded HCEs evoked a mild but significant increase in the intracellular H_2_S levels (*p* < 0.001). 

In Figure 5, the results of the fluorometric assay of brimonidine hybrids are graphically represented. All the compounds led to a significant release of hydrogen sulfide (*p* < 0.001), except for the ADT-OH conjugated hybrid (**3b**). Among the compounds **3a**–**3d**, the molecular hybrid **3d** (brimonidine–HPI) showed the highest increase in fluorescence.

Analyzing the data from the amperometric and the fluorometric assays, the molecular hybrids synthetized by coupling HBTA **9** and HPI **10** with antiglaucoma drugs (**1**–**3**) released a higher amount of H_2_S in aqueous buffer as well as in the cells, compared to the molecular hybrids of TBZ **7** and ADT-OH **8**. Furthermore, evaluating the influence of the antiglaucoma drugs in the release of H_2_S, betaxolol hybrids demonstrated a weak generation of sulfide when compared to brinzolamide and brimonidine derivatives. 

Therefore, compounds **1c**, **1d** and **3d** showed the best releasing profiles, leading to an enhanced H_2_S production. Besides the amount of the gasotransmitter produced, the H_2_S-releasing kinetic also influences biological activity. The amperometic assay demonstrated that these hybrids had a progressive and long-lasting release of H_2_S in the presence of L-cysteine, acting as smart donors. These features are considered as indispensable for the potential clinical application of H_2_S donors, since they avoid the side effects related to a fast release (typical of the sulfide and hydrosulfide salts) and also mimic the endogenous H_2_S production.

## 4. Experimental Section

### 4.1. Materials and Methods

Brinzolamide and brimonidine were purchased from Abcr (Karlsruhe, Germany); betaxolol was purchased from Carbosynth (Compton, UK). All reagents, solvents and other chemicals were commercial products obtained from Merck (Darmstadt, Germany). Melting points, determined using a Buchi Melting Point B-540 instrument (Flawil, Switzerland), are uncorrected and represent values obtained on recrystallized or chromatographically purified material. Spectra of ^1^H and ^13^C NMR were recorded on a Bruker Advanced 400 MHz spectrometer (Billerica, MA, USA). Spectra of brinzolamide and brimonidine derivatives were recorded in DMSO-*d*_6_. Spectra of betaxolol hybrids were recorded in CD_3_OD and CDCl_3_ (compound **2d**). Chemical shifts are reported in ppm. The following abbreviations are used to describe peak patterns when appropriate: s (singlet), d (doublet), t (triplet), m (multiplet), q (quartet), qt (quintet), dd (doublet of doublet), td (triplet of doublets), bs (broad singlet). Mass spectra of the intermediates and final products were recorded on an LTQ-XL mass spectrometer equipped with a HESI ion source (Thermo Fisher Scientific, Waltham, MA, USA). All reactions were followed by thin-layer chromatography, carried out on Merck silica gel 60 F_254_ plates with a fluorescent indicator, and the plates were visualized with UV light (254 nm). Preparative chromatographic purifications were performed using a silica gel column (Kieselgel 60). Solutions were concentrated with a Buchi R-114 rotary evaporator at low pressure. 

### 4.2. Synthesis of Compounds ***1a***–***1d***

#### 4.2.1. 2-(4-Carbamothioylphenoxy)-N-ethyl-N-(2-(3-methoxypropyl)-1,1-dioxido-6-sulfamoyl-3,4-dihydro-2H-thieno[3,2-e][1,2]thiazin-4-yl)acetamide (Brinzolamide–TBZ, Compound **1a**)

Commercially available brinzolamide **1** (1.00 g; 2.61 mmol) was solubilized in DMF (30 mL) and condensed with the derivative **7b** (0.551 g; 2.61 mmol), via TBTU (1.00 g; 3.13 mmol) and HOBt (0.423 g; 3.13 mmol) in the presence of N,N-diisopropylethylamine (0.910 mL; 5.22 mmol). The mixture was stirred at room temperature for 12 h. The solvent was evaporated and the residue was then purified by silica gel open chromatography using dichloromethane/methanol as eluent (9:1 *v*/*v*). The compound **1a** was then isolated as a yellowish oil. Yield: 0.628 g; 41.7%. 

^1^H NMR (400 MHz, DMSO-*d_6_*) δ: 9.59 (bs, 2H), 9.28 (bs, 2H), 7.88 (d, *J* = 8.5 Hz, 2H), 7.65 (s, 1H), 6.80 (d, *J* = 12.5 Hz, 2H), 4.40 (s, 2H), 4.12–4.10 (m, 1H), 3.87–3.85 (m, 2H), 3.39–3.35 (m, 3H), 3.23 (s, 3H), 3.17–3.15 (m, 1H), 2.83–2.77 (m, 2H), 1.83–1.80 (m, 2H), 1.08 (t, *J* = 7.0 Hz, 3H); ^13^C NMR (101 MHz, DMSO-*d_6_*) δ: 199.13, 173.05, 161.67, 151.95, 131.60, 129.62, 128.51, 127.69, 124.95, 119.56, 113.99, 110.13, 69.31, 69.10, 58.40, 54.23, 49.01, 45.84, 29.02. ESI-MS *m*/*z* [M+H]^+^ calculated for C_21_H_28_N_4_O_7_S_4_ 576.73, found = 577.2. 

#### 4.2.2. N-Ethyl-N-(2-(3-methoxypropyl)-1,1-dioxido-6-sulfamoyl-3,4-dihydro-2H-thieno[3,2-e][1,2]thiazin-4-yl)-2-(4-(3-thioxo-3H-1,2-dithiol-5-yl)phenoxy)acetamide (Brinzolamide–ADTOH, Compound **1b**)

Following the synthetic procedure described above for **1a**, compound **1b** was synthetized starting from brinzolamide **1** (1.00 g; 2.61 mmol) and the derivative **8b** (0.742 g; 2.61 mmol), and isolated as an orange solid. Yield: 0.889 g; 52.4%. Mp: 154.1–155.6 °C. 

^1^H NMR (400 MHz, DMSO-*d_6_*) δ: 8.2 (bs, 2H), 7.82 (d, *J* = 8.5 Hz, 2H), 7.73 (s, 1H), 6.93 (d, *J* = 12.5 Hz, 2H), 5.76 (s, 1H), 4.47 (s, 2H), 3.94–3.93 (1H, m), 3.63–3.60 (m, 2H), 3.39–3.35 (m, 3H), 3.23 (s, 3H), 3.18–3.14 (m, 1H), 2.89–2.87 (m, 2H), 1.84–1.80 (m, 2H), 1.13 (t, *J* = 7.4 Hz, 3H); ^13^C NMR (101 MHz, DMSO-*d_6_*) δ: 215.13, 174.47, 172.87, 162.56, 134.48, 129.21, 128.50, 127.80, 124.98, 123.84, 119.60, 116.05, 110.08, 69.30, 69.13, 58.41, 54.05, 48.78, 45.85, 29.01, 17.20. ESI-MS *m*/*z* [M+H]^+^ calculated for C_23_H_27_N_3_O_7_S_6_ 649.87, found = 650.1.

#### 4.2.3. Ethyl 4-(2-(ethyl(2-(3-methoxypropyl)-1,1-dioxido-6-sulfamoyl-3,4-dihydro-2H-thieno[3,2-e][1,2]thiazin-4-yl)amino)-2-oxoethoxy)benzodithioate (Brinzolamide–HBTA, Compound **1c**)

Following the synthetic procedure described above for **1a**, compound **1c** was synthetized starting from brinzolamide **1** (1.00 g; 2.61 mmol) and the derivative **9b** (0.670 g; 2.61 mmol), and isolated as a pink solid. Yield: 1.280 g; 78.9%. Mp: 174.0–175.6 °C. 

^1^H NMR (400 MHz, DMSO-*d_6_*) δ: 8.19 (bs, 2H), 7.95 (d, *J* = 8.8 Hz, 2H), 7.59 (s, 1H), 6.85 (d, *J* = 12.5 Hz, 2H), 4.43 (s, 2H), 3.83–3.80 (m, 1H), 3.62–3.58 (m, 2H), 3.40–3.34 (m, 5H), 3.21 (s, 3H), 3.15–3.13 (m, 1H), 2.70–2.65 (m, 2H), 1.80–1.78 (m, 2H), 1.31 (t, *J* = 6.1 Hz, 3H), 1.05 (t, *J* = 7.0 Hz, 3H); ^13^C NMR (101 MHz, DMSO-*d_6_*) δ: 225.99, 172.69, 163.44, 155.90, 137.46, 128.79, 128.48, 114.81, 110.25, 69.29, 69.15, 58.38, 54.05, 45.82, 43.25, 31.01, 29.02, 18.54, 17.18, 12.84. ESI-MS *m*/*z* [M+H]^+^ calculated for C_23_H_31_N_3_O_7_S_5_ 621.83, found = 622.1.

#### 4.2.4. 4-Isothiocyanatophenyl 4-(ethyl(2-(3-methoxypropyl)-1,1-dioxido-6-sulfamoyl-3,4-dihydro-2H-thieno[3,2-e][1,2]thiazin-4-yl)amino)-4-oxobutanoate (Brinzolamide–HPI, Compound **1d**)

The synthesis of compound **1d** occurs in two steps. The first reaction was performed in acetonitrile (20 mL) as solvent, with azeotropic elimination of water from the system [37]. Succinic anhydride (0.287 g; 2.87 mmol) was added to a solution of brinzolamide **1** (1.00 g; 2.61 mmol) and the mixture was stirred overnight at reflux. The solvent was evaporated under reduced pressure and the crude residue was then purified by silica gel open chromatography (dichloromethane/methanol 9:1 *v*/*v*) to obtain the acid derivative **4** as colorless oil. Yield: 0.891 g; 70.6%. ESI-MS *m*/*z* [M+H]^+^ calculated for C_16_H_25_N_3_O_8_S_3_ 483.58, found = 484.4.

In the second step, the synthetized intermediate **4** (1.00 g; 2.07 mmol) was solubilized in DMF (30 mL) and condensed with HPI **10** (0.313 g; 2.07 mmol), by TBTU coupling (0.796 g; 2.48 mmol), in presence of HOBt (0.335 g; 2.48 mmol) and N,N-diisopropylethylamine (0.721 mL; 4.14 mmol). The mixture was stirred at room temperature for 12 h. The solvent was evaporated and the crude material was purified by silica gel open chromatography using ethyl acetate/diethyl ether as eluent (9,5:0,5 *v*/*v*). Then, the compound **1d** was isolated by crystallization from n-hexane as a white solid. Yield: 0.572 g; 44.8%. Mp: 95.5–96.4 °C.

^1^H NMR (400 MHz, DMSO-*d_6_*) δ: 8.00 (bs, 2H), 7.46 (d, *J* = 8.5 Hz, 2H), 7.25 (s, 1H), 7.17 (d, *J* = 12.5 Hz, 2H), 4.00–3.97 (m, 1H), 3.61–3.58 (m, 2H), 3.47–3.42 (m, 1H), 3.37 (t, *J* = 6.1 Hz, 2H), 3.22 (s, 3H), 3.21–3.18 (m, 1H), 2.83–2.77 (m, 6H), 1.85–1.81 (m, 2H), 0.99 (t, *J* = 7.1 Hz, 3H); ^13^C NMR (101 MHz, DMSO-*d_6_*) δ: 171.65, 150.00, 149.12, 142.83, 139.95, 134.15, 129.85, 127.84, 127.70, 127.63, 123.70, 69.25, 58.40, 45.74, 31.42, 29.60, 29.14, 28.60, 22.53, 14.80, 14.43. ESI-MS *m*/*z* [M+H]^+^ calculated for C_23_H_28_N_4_O_8_S_4_ 616.75, found = 617.1.

### 4.3. Synthesis of Compounds ***2a***–***2d***

#### 4.3.1. 2-(4-Carbamothioylphenoxy)-N-(3-(4-(2-(cyclopropylmethoxy ethyl) phenoxy)-2-hydroxypropyl)-N-isopropylacetamide (Betaxolol–TBZ, Compound **2a**)

Compound **2a** was obtained according to the procedure reported above for **1a**, starting from betaxolol **2** (1.00 g; 3.25 mmol) and the intermediate **7b** (0.686 g; 3.25 mmol), and isolated as a yellow solid. Yield: 0.511 g; 31.4%. Mp: 73.1–74.5 °C.

The analysis of ^1^H, ^13^C and bidimensional NMR spectra showed that betaxolol hybrids **2a**–**2c** are a mixture of *cis/trans* isomers (Figure 6). As reported in details in the literature, unsymmetrically N,N-disubstituted amides are characterized by a hindered rotation around the C(O)-N bond but the energy difference between the two conformations is small and the molecules are a combination of *cis/trans* isomers [38,39,40,41]. The rate of conversion between conformational isomers of betaxolol hybrids is sufficiently slow to allow a chemical shift difference of signals arising from *cis* and *trans* isomers.

^1^H NMR (400 MHz, CD_3_OD) δ: 7.95 (d, *J* = 4.5 Hz, 2H), 7.92 (d, *J* = 4.6 Hz, 2H), 7.17 (d, *J* = 4.3 Hz, 2H), 7.15 (d, *J* = 6.4 Hz, 2H), 6.99 (d, *J* = 4.4 Hz, 2H), 6.97 (d, *J* = 4.5 Hz, 2H), 6.89 (d, *J* = 8.5 Hz, 2H), 6.83 (d, *J* = 8.6 Hz, 2H), 5.19 (d, *J* = 15 Hz, 1H), 4.98 (s, 2H), 4.90 (d, *J* = 14.9 Hz, 1H), 4.36–4.32 (m, 1H), 4.25–4.17 (m, 2H), 4.04–4.02 (m, 1H), 3.99–3.97 (m, 1H), 3.92–3.94 (m, 1H), 3.87–3.85 (m, 1H), 3.66–3.64 (m, 2H), 3.61 (d, *J* = 5.2 Hz, 1H), 3.59 (d, *J* = 5.2 Hz, 1H) 3.55–3.54 (m, 1H), 3.39 (d, *J* = 7.0 Hz, 1H), 3.37 (d, *J* = 6.9 Hz, 1H), 3.31 (d, *J* = 3.6 Hz, 2H), 3.30 (d, *J* = 6.9 Hz, 2H), 2.80 (td, *J* = 7.1 Hz, 2.0 Hz, 2H) 1.34 (t, *J* = 6.1 Hz, 3H), 1.32 (d, *J* = 6.8 Hz, 3H), 1.28 (d, *J* = 6.6 Hz, 3H), 1.03 -1.00 (m, 1H), 0.49–0.51 (m, 2H), 0.19–0.17 (m, 2H); ^13^C NMR (101 MHz, CD_3_OD) δ: 202.53, 202.40, 171.08, 170.79, 162.86, 158.71, 158.56, 133.97, 133.56, 132.97, 132.64, 131.00, 130.92, 130.59, 130.43, 115.51, 115.45, 115.07, 115.02, 76.60, 72.84, 72.80, 71.44, 71.27, 70.18, 70.12, 67.85, 67.71, 50.48, 49.84, 48.14, 45.78, 36.31, 21.67, 21.28, 20.68, 20.16, 11.40, 3.41. ESI-MS *m*/*z* [M+H]^+^ calculated for C_27_H_36_N_2_O_5_S 500.65, found = 501.4.

#### 4.3.2. N-(3-(4-(2-(Cyclopropylmethoxy)ethyl)phenoxy)-2-hydroxypropyl)-N-isopropyl-2-(4-(3-thioxo-3H-1,2-dithiol-5-yl)phenoxy)acetamide (Betaxolol–ADTOH, Compound **2b**)

Compound **2b** was obtained according to the procedure reported above for **1a**, starting from betaxolol **2** (1.00 g; 3.25 mmol) and the intermediate **8b** (0.924 g; 3.25 mmol), and isolated as an orange oil. Yield: 0.699 g; 37.5%.

^1^H NMR (400 MHz, CD_3_OD) δ: 7.73 (d, *J* = 4.5 Hz, 2H), 7.48 (d, *J* = 20.4 Hz, 2H), 7.15 (d, *J* = 8.5 Hz, 2H), 7.11–7.08 (m, 4H), 7.07 (d, *J* = 4.5 Hz, 2H), 6.88 (d, *J* = 4.5 Hz, 2H), 6.78 (d, *J* = 4.6 Hz, 2H), 5.50 (s, 1H), 5.26 (d, *J* = 15.1 Hz, 1H), 5.02 (s, 2H), 4.95 (d, *J* = 15 Hz, 1H), 4.36–4.29 (m, 1H), 4.26–4.15 (1H, m), 4.04–3.95 (1H, m), 3.88–3.78 (1H, m), 3.63 (t, *J* = 7.1 Hz, 2H), 3.55 (d, *J* = 5.8 Hz, 1H), 3.50 (q, *J* = 7.0 Hz, 2H), 3.38 (d, *J* = 6.8 Hz, 1H), 3.29 (d, *J* = 2.1 Hz, 2H) 3.27 (d, *J* = 2.1 Hz, 2H), 2.81–2.78 (m, 2H) 1.35–1.28 (m, 3H), 1.19 (t, *J* = 7.0 Hz, 3H), 1.04–1.01 (1H, m), 0.53–0.49 (m, 2H), 0.21–0.17 (m, 2H); ^13^C NMR (101 MHz, CD_3_OD) δ: 217.21, 174.88, 174.58, 170.83, 170.53, 163.41, 162.93, 158.70, 158.56, 135.81, 135.67, 132.99, 132.65, 130.99, 130.88, 129.85, 129.66, 126.21, 126.14, 125.80, 116.94, 116.88, 115.52, 115.39, 76.59, 72.81, 71.60, 71.25, 70.15, 70.03, 67.86, 67.75, 66.91, 50.50, 49.84, 48.06, 45.63, 38.88, 36.61, 30.89, 30.75, 21.69, 21.27, 20.68, 20.17, 15.43, 11.40, 3.43. ESI-MS *m*/*z* [M+H]^+^ calculated for C_29_H_35_NO_5_S_3_ 573.79, found = 574.31

#### 4.3.3. Ethyl 4-(2-((3-(4-(2-(cyclopropylmethoxy)ethyl)phenoxy)-2-hydroxypropyl)(isopropyl)amino)-2-oxoethoxy)benzodithioate (Betaxolol–HBTA, Compound **2c**)

Compound **2c** was obtained according to the procedure reported above for **1a**, starting from betaxolol **2** (1.00 g; 3.25 mmol) and the intermediate **9b** (0.833 g; 3.25 mmol), and isolated as a pink oil. Yield: 0.816 g; 46.0%.

^1^H NMR (400 MHz, CD_3_OD) δ: 8.07 (d, *J* = 4.8 Hz, 2H), 8.04 (d, *J* = 4.8 Hz, 2H), 7.17 (d, *J* = 6.5 Hz, 2H), 7.12 (d, *J* = 6.6 Hz, 2H), 6.98 (d, *J* = 4.8 Hz, 2H), 6.96 (d, *J* = 4.8 Hz, 2H), 6.89 (d, *J* = 4.5 Hz, 2H), 6.82 (d, *J* = 4.4 Hz, 2H), 5.22 (d, *J* = 14.9 Hz, 1H), 4.99 (s, 2H), 4.92 (d, *J* = 14.9 Hz, 1H), 4.38–4.31 (m, 1H), 4.24–4.18 (m, 3H), 4.05–3.95 (m, 2H), 3.94–3.91 (m, 1H), 3.86–3.81 (m, 1H), 3.61 (t, *J* = 7.1 Hz, 2H), 3.58 (d, *J* = 5.3 Hz, 1H), 3.54–3.52 (m, 1H), 3.42–3.36 (m, 4H), 3.30 (d, *J* = 3.1 Hz, 2H), 3.28 (d, *J* = 3.0 Hz, 2H), 2.82 -2.78 (m, 2H), 1.39 (td, *J* = 7.4, 2.0 Hz, 3H), 1.30–1.35 (m, 6H), 1.28 (d, *J* = 6.6 Hz, 3H), 1.06–0.99 (m, 1H), 0.53–0.49 (m, 2H), 0.21–0.17 (m, 2H); ^13^C NMR (101 MHz, CD_3_OD) δ: 202.92, 202.59, 170.88, 170.60, 163.94, 163.53, 158.69, 158.53, 140.03, 139.74, 132.94, 132.55, 130.99, 130.90, 129.82, 129.56, 115.50, 115.43, 115.32, 76.58, 72.85, 72.80, 71.43, 71.21, 70.14, 70.07, 67.81, 67.73, 50.47, 49.81, 48.08, 45.72, 36.31, 31.93, 31.85, 21.67, 21.27, 20.69, 20.18, 12.87, 11.40, 3.42. ESI-MS *m*/*z* [M+H]^+^ calculated for C_29_H_39_NO_5_S_2_ 545.75, found = 546.32

#### 4.3.4. 1-(4-(2-(Cyclopropylmethoxy)ethyl)phenoxy)-3-(isopropylamino)propan-2-yl (4-isothiocyanatophenyl) Succinate (Betaxolol-HPI, Compound **2d**)

Betaxolol **2** (1.00 g; 3.25 mmol), solubilized in anhydrous dichloromethane (30 mL), was treated with a catalytic amount of DMAP (0.040 g; 0.32 mmol). The solution was cooled to 0 °C, and succinic anhydride (0.488 g; 4.87 mmol) was added with the mixture being stirred at room temperature for 6h. The solvent was concentrated in vacuo and the resulting hemisuccinated ester **5** was isolated by silica gel open chromatography (dichloromethane/methanol 9:1 *v*/*v*) as an oil. Yield: 0.539 g; 40.7%. ESI-MS *m*/*z* [M+H]^+^ calculated for C_22_H_33_NO_6_ 407.50, found = 408.6.

The intermediate **5** (1.00 g; 2.45 mmol) was linked to HPI **10** (0.370 g; 2.45 mmol) using EDAC·HCl (0.703 g; 3.67 mmol) and DMAP (0.448 g; 3.67 mmol) as coupling agents in anhydrous THF (20 mL), for 12 h at room temperature. The solvent was removed to obtain the crude product. The residue was loaded on a silica gel open column and eluted with dichloromethane/ethyl acetate (9.5:0.5 *v*/*v*). The combined and evaporated fractions produced compound **2d** as a colorless oil. Yield: 0.395 g; 29.8%.

^1^H NMR (400 MHz, CDCl_3_) δ: 7.22 (d, *J* = 4.5 Hz, 2H), 7.13 (d, *J* = 8.6 Hz, 2H), 7.09 (d, *J* = 4.5 Hz, 2H), 6.83 (d, *J* = 4.7 Hz, 2H), 4.18–4.11 (m, 1H), 4.04–3.93 (m, 2H) 3.83–3.78 (m, 1H), 3.61 (t, *J* = 7.4 Hz, 2H), 3.46 (dd, *J* = 14.7, 1.8 Hz, 2H), 3.28 (d, *J* = 6.9 Hz, 2H), 2.95–2.80 (m, 6H), 2.00 (bs, 1H) 1.29 (d, *J* = 6.7 Hz, 3H), 1.23 (d, *J* = 6.5 Hz, 3H), 1.07–1.03 (m, 1H), 0.55–0.50 (m, 2H), 0.21–0.17 (m, 2H); ^13^C NMR (101 MHz, CDCl_3_) δ: 173.76, 171.53, 157.01, 149.49, 136.02, 131.67, 130.04, 128.93, 126.86, 123.04, 114.40, 75.78, 72.32, 71.93, 69.70, 49.18, 46.36, 35.60, 29.61, 28.54, 21.23, 20.83, 10.76, 3.12. ESI-MS *m*/*z* [M+H]^+^ calculated for C_29_H_36_N_2_O_6_S 540.67, found = 541.3.

### 4.4. Synthesis of Compounds ***3a***–***3d***

#### 4.4.1. 4-Carbamothioylphenyl 4-((5-bromoquinoxalin-6-yl)(4,5-dihydro-1H-imidazol-2-yl)amino)-4-oxobutanoate (Brimonidine–TBZ, Compound **3a**)

A solution of succinic anhydride (0.376 g; 3.76 mmol) and DMAP (0.041 g; 0.34 mmol) solubilized in DMF (10 mL) was added to brimonidine **3** (1.00 g; 3.42 mmol) in anhydrous DMF (5 mL), under a nitrogen atmosphere, at room temperature, and the mixture was stirred overnight. Subsequently, to the obtained intermediate **6** (not isolated), TBZ **7** (0.524 g; 3.42 mmol), EDAC·HCl (0.983 g; 5.13 mmol) and DMAP (0.627 g; 5.13 mmol) were added. The reaction mixture was stirred at room temperature for 12 h. The solvent was removed in vacuo and the residue was purified by column chromatography on silica gel (ethyl acetate/dichloromethane 8:2 *v*/*v*). The yellow solid **3a** was obtained by recrystallization with diethyl ether. Yield: 0.610 g; 33.8%. Mp: 121.8–123.4 °C.

^1^H NMR (400 MHz, DMSO-*d_6_*) δ: 9.88 (bs, 1H), 9.52 (bs, 1H), 8.95 (s, 1H), 8.83 (s, 1H), 7.99 (d, *J* = 8.5 Hz, 1H), 7.94 (d, *J* = 8.2 Hz, 2H), 7.59 (d, *J* = 8.9 Hz, 1H), 7.15 (d, *J* = 8.5 Hz, 2H), 7.11 (bs, 1H), 3.94–3.93 (m, 2H), 3.54–3.52 (m, 2H), 3.37–3.35 (m, 2H), 2.94–2.92 (m, 2H); ^13^C NMR (101 MHz, DMSO-*d_6_*) δ: 199.52, 171.79, 171.69, 153.24, 150.89, 149.89, 146.04, 143.89, 141.83, 140.47,137.42, 129.36, 129.23, 128.46, 121.60, 115.38, 44.07, 38.93, 32.50, 29.24. ESI-MS *m*/*z* [M+H]^+^ calculated for C_22_H_19_BrN_6_O_3_S 526.04, found = 527.2.

#### 4.4.2. 4-(3-Thioxo-3H-1,2-dithiol-5-yl)phenyl 4-((5-bromoquinoxalin-6-yl)(4,5-dihydro-1H-imidazol-2-yl)amino)-4-oxobutanoate (Brimonidine–ADTOH, Compound **3b**)

Compound **3b** was synthetized following the synthetic route applied for the synthesis of **3a**. Brimonidine **3** (1.00 g; 3.42 mmol) was linked to ADT-OH **8** (0.774 g; 3.42 mmol). The compound **3b** was then isolated as an orange solid. Yield: 0.700 g; 34.1%. Mp 128.1–129.5 °C.

^1^H NMR (400 MHz, DMSO-*d_6_*) δ: 8.94 (s, 1H), 8.83 (s, 1H), 7.99 (d, *J* = 6.0 Hz, 1H), 7.83 (d, *J* = 10 Hz, 2H), 7.59 (d, *J* = 8.6 Hz, 1H), 7.29 (d, *J* = 8.1 Hz, 2H), 7.12 (bs, 1H), 5.70 (s, 1H), 3.94–3.91 (m, 2H), 3.54–3.52 (m, 2H), 3.38–3.35 (m, 2H), 2.95–2.93 (m, 2H); ^13^C NMR (101 MHz, DMSO-*d_6_*) δ: 215.96, 173.24, 171.79, 171.62, 154.07, 150.89, 149.90, 146.05, 143.91, 141.83, 140.48, 136.23, 129.31, 129.17, 128.46, 123.51, 123.47, 115.39, 44.08, 38.94, 32.54, 29.30. ESI-MS *m*/*z* [M+H]^+^ calculated for C_24_H_18_BrN_5_O_3_S_3_ 598.98, found = 600.1.

#### 4.4.3. 4-((Ethylthio)carbonothioyl)phenyl 4-((5-bromoquinoxalin-6-yl)(4,5-dihydro-1H-imidazol-2-yl)amino)-4-oxobutanoate (Brimonidine–HBTA, Compound **3c**)

Compound **3c** was synthetized following the synthetic route applied for the synthesis of **3a**. Brimonidine **3** (1.00 g; 3.42 mmol) was linked to HBTA **9** (0.678 g; 3.42 mmol). Compound **3c** was then isolated as a pink solid. Yield: 0.869 g; 44.4%. Mp: 160.5–162.1 °C.

^1^H NMR (400 MHz, DMSO-*d_6_*) δ: 8.94 (s, 1H), 8.83 (s, 1H), 8.02 (d, *J* = 8.5 Hz, 2H), 7.99 (d, *J* = 8.9 Hz, 1H), 7.60 (d, *J* = 8.6 Hz, 1H), 7.25 (d, *J* = 7.6 Hz, 2H), 7.11 (bs, 1H), 3.93 (t, *J* = 7.7 Hz, 2H), 3.55 (t, *J* = 6.0 Hz, 2H), 3.41–3.35 (m, 4H), 2.96–2.90 (m, 2H), 1.35 (t, *J* = 7.3 Hz, 3H); ^13^C NMR (101 MHz, DMSO-*d_6_*) δ: 225.4, 171.32, 171.06, 154.18, 150.41, 149.42, 145.56, 143.42, 141.83, 141.36, 140.00, 128.83, 127.98, 127.89, 122.00, 114.90, 43.59, 38.45, 32.02, 31.14, 28.31, 12.13. ESI-MS *m*/*z* [M+H]^+^ calculated for C_24_H_22_BrN_5_O_3_S_2_ 571.03, found = 572.2

#### 4.4.4. 4-Isothiocyanatophenyl 4-((5-bromoquinoxalin-6-yl)(4,5-dihydro-1H-imidazol-2-yl)amino)-4-oxobutanoate (Brimonidine–HPI, Compound **3d**)

Compound **3d** was synthetized following the synthetic route applied for the synthesis of **3a**. Brimonidine **3** (1.00 g; 3.42 mmol) was linked to HPI **10** (0.517 g; 3.42 mmol). The compound **3d** was then insolated as a pale yellow solid. Yield: 1.146 g; 63.8%. Mp: 159.0–160.5 °C.

^1^H NMR (400 MHz, DMSO-*d_6_*) δ: 8.94 (s, 1H), 8.83 (s, 1H), 7.98 (d, *J* = 8.9 Hz, 1H), 7.58 (d, *J* = 8.9 Hz, 1H), 7.49 (d, *J* = 8.7 Hz, 2H), 7.19 (d, *J* = 8.7 Hz, 2H), 7.09 (bs, 1H), 3.92 (t, *J* = 7.8 Hz, 2H), 3.52 (t, *J* = 6.3Hz, 2H), 3.36 (t, *J* = 7.8, 2H), 2.90 (t, *J* = 6.2 Hz, 2H); ^13^C NMR (101 MHz, DMSO-*d_6_*) δ: 171.26, 150.41, 149.42, 145.57, 143.42, 141.36, 140.00, 133.66, 128.83, 127.98, 127.39, 127.21, 123.28, 121.60, 116.33, 114.91, 43.59, 38.45, 32.05, 28.74. ESI-MS *m*/*z* [M+H]^+^ calculated C_22_H_17_BrN_6_O_3_S 524.03, found = 525.1.

### 4.5. Synthesis of Intermediates ***7a***–***9a***

#### 4.5.1. Tert-butyl 2-(4-carbamothioylphenoxy)acetate (**7a**)

In a two-neck flask, sodium hydride (60% dispersion in mineral oil, 0.261 g; 6.53 mmol) was suspended in DMF (10 mL) and the suspension was stirred and cooled to 0 °C. A solution of TBZ **7** (1.00 g; 6.53 mmol) in DMF (2 mL) was added dropwise. After 10 min a solution of tert-butyl bromoacetate (1.16 mL; 7.84 mmol) in DMF (2 mL) was added dropwise. The mixture was stirred at room temperature for 12 h. The solution was concentrated in vacuo and the crude residue was purified by silica gel open chromatography (dichloromethane as eluent) to produce intermediate **7a** as a yellowish solid. Yield: 1.427 g; 81.7%. ESI-MS *m*/*z* [M+H]^+^ calculated C_13_H_17_NO_3_S 267.34, found = 268.6.

#### 4.5.2. Tert-butyl 2-(4-(3-thioxo-3H-1,2-dithiol-5-yl)phenoxy)acetate (**8a**)

Compound **8a** was synthetized from ADT-OH **8** (1.00 g; 4.42 mmol) and tert-butyl bromoacetate (0.783 mL; 5.30 mmol) in the presence of NaH (0.177 g; 4.42 mmol), following the procedure adopted for the synthesis of **7a**, and isolated as a brown solid. Yield: 0.957 g; 63.6%. ESI-MS *m*/*z* [M+H]^+^ calculated C_15_H_16_O_3_S_3_ 340.48, found = 342.0.

#### 4.5.3. Tert-butyl 2-(4-((ethylthio)carbonothioyl)phenoxy)acetate (**9a**)

Compound **9a** was synthetized from HBTA (1.00 g; 5.04 mmol) and tert-butyl bromoacetate (0.893 mL; 6.05 mmol) in the presence of NaH (0.202 g; 5.04 mmol), following the procedure adopted for the synthesis of **7a**, and isolated as a pink solid. Yield: 0.821 g; 52.1%. ESI-MS *m*/*z* [M+H]^+^ calculated C_15_H_20_O_3_S_2_ 312.45, found = 313.8.

### 4.6. General Procedure for the Synthesis of Intermediates ***7b***–***9b***

Intermediates **7a**–**9a** were dissolved in a 10% (*v*/*v*) TFA solution in anhydrous dichloromethane (10 mL) and stirred at room temperature until the compound was completely deprotected. Solvent was then removed by reduced pressure distillation and the compounds **7b**–**9b** were obtained by recrystallization with diethyl ether.

#### 4.6.1. 2-(4-Carbamothioylphenoxy)acetic acid (**7b**)

Synthetized from intermediate **7a** and isolated as a yellowish solid. Yield: 91.1%. ESI-MS m/z [M+H]^+^ calculated C_9_H_9_NO_3_S 211.24, found = 212.1.

#### 4.6.2. 2-(4-(3-Thioxo-3H-1,2-dithiol-5-yl)phenoxy)acetic acid (**8b**)

Synthetized from intermediate **8a** and isolated as an orange solid. Yield: 95.4%. ESI-MS m/z [M+H]^+^ calculated C_11_H_8_O_3_S_3_ 284,37, found = 285.2.

#### 4.6.3. 2-(4-((Ethylthio)carbonothioyl)phenoxy)acetic acid (**9b**)

Synthetized from intermediate **9a** and isolated as a pink solid. Yield: 93.2%. ESI-MS m/z [M+H]^+^ calculated C_11_H_12_O_3_S_2_, found 256.34 = 257.9.

### 4.7. Amperometric Determination of H_2_S Release

The H_2_S-releasing properties of compounds **1a**–**1d**, **2a**–**2d** and **3a**–**3d** were evaluated by amperometry, through an Apollo-4000 Free Radical Analyzer (World Precision Instrument, WPI) detector and H_2_S-selective minielectrodes (ISO-H_2_S-2, WPI) endowed with gas-permeable membranes [25]. The experiments were carried out at room temperature. Following the instructions of the manufacturer, a “PBS buffer 10×” was prepared (NaH_2_PO_4_·H_2_O, 1.28 g; Na_2_HPO_4_·12H_2_O, 5.97 g; and NaCl, 43.88 g in 500 mL of H_2_O) and stocked at 4 °C. Immediately before the experiments, the “PBS buffer 10×” was diluted in distilled water (1:10) to obtain the assay buffer (AB); pH was adjusted to 7.4. The H_2_S-selective minielectrode was equilibrated in 2 mL of the AB until the recovery of a stable baseline. Then, 20 μL of a dimethyl sulfoxide (DMSO) solution of the tested compounds were added (final concentration, 100 μM; final concentration of DMSO in the AB, 1%). The generation of H_2_S was observed for 30 min. When required by the experimental protocol, L-cysteine 4 mM was added, before the H_2_S-releasing molecule. The relationship between the amperometric currents (recorded in pA) and the corresponding concentrations of H_2_S was determined by calibration curves with increasing concentrations of NaHS (1 μM, 3 μM and 7 μM) at pH 4.0. The curves relative to the progressive increase of H_2_S vs. time, following the incubation of the tested compounds, were analyzed by a fitting curve using the software GraphPad Prism 6.0. The parameter of C_max_ (the highest concentration of H_2_S obtained during the recording time) and TCM_50_ (time required to reach a concentration = ½ C_max_) were calculated and expressed as mean ± standard error from five different experiments. ANOVA and Student’s *t*-test were selected as statistical analysis, *p* < 0.05 was considered representative of significant statistical differences.

### 4.8. In Vitro Evaluation

#### 4.8.1. Cell Culture

Human primary corneal epithelial cells (HCEs) were grown in corneal epithelial cell basal media supplemented with corneal epithelial cell growth kit components and 1% of 100 units/mL penicillin and 100 mg/mL streptomycin (Sigma Aldrich) in a tissue culture flask at 37 °C in a humidified atmosphere and 5% CO_2_. HCEs were cultured up to about 90% confluence and 24 h before the experiment; the cells were seeded onto a 96-well black plate, clear bottom pre-coated with gelatin 1% (from porcine skin, Sigma Aldrich), at density of 72 × 10^3^ per well. Cells were split 1:2 twice a week and used until passage 18.

#### 4.8.2. Evaluation of H_2_S Release on HCEs

After 24 h to allow cell attachment, the medium was replaced and cells were incubated for 30 min in the buffer standard (HEPES, 20 mM; NaCl, 120 mM; KCl, 2 mM; CaCl_2_·2H_2_O, 2 mM; MgCl_2_·6H_2_O, 1 mM; Glucose, 5 mM; and pH, 7.4, at room temperature) containing the fluorescent dye WSP-1 (Washington State Probe-1, 1,3′-methoxy-3-oxo-3H-spiro[isobenzofuran-1,9′-xanthen]-6′yl 2-(pyridin-2-yldisulfanyl) benzoate, Cayman Chemical) at the concentration of 100 μM [33,42]. Then, the supernatant was removed and replaced with a solution of the tested compounds or diallyl disulfide (DADS) as a known H_2_S donor in buffer standard [29]. When WSP-1 reacts with H_2_S, it releases a fluorophore detectable with a spectrofluorometer at excitation and emission wavelengths of 465–515 nm [25,28,33]. The increasing of fluorescence (expressed as fluorescence index = FI) was monitored after 30 min, using a spectrofluorometer (EnSpire, Perkin Elmer).

#### 4.8.3. Statistical Analysis

Experimental data were analyzed by a computer fitting procedure (software: GraphPad Prism 6.0) and expressed as mean ± standard error; three different experiments were performed, each carried out in three replicates. ANOVA and Student’s *t*-test were selected as statistical analyses; when required, the Bonferroni post hoc test was used. *p* < 0.05 was considered as representative of significant statistical differences.

## 5. Conclusions

For most forms of glaucoma, including normotensive glaucoma, pharmacological treatment is currently based on IOP control through topical medications. However, the last topical agent for glaucoma therapy approved by the Food and Drug Administration (FDA) dates back to more than 20 years ago [43]. Therefore, with the increasing prevalence of glaucoma worldwide, the exigency of new therapies is emerging.

In this work, we synthetized and characterized new molecular hybrids between currently available drugs for glaucoma therapy and H_2_S-releasing compounds to improve the efficacy of antiglaucoma medications and reduce side effects.

We synthetized hybrid derivatives of brinzolamide (carbonic anhydrase inhibitor; compounds **1a**–**1d**), betaxolol (β-blocker; compounds **2a**–**2d**) and brimonidine (α_2_-adrenergic agonist; compounds **3a**–**3d**).

The new molecular entities were tested for their H_2_S-releasing properties via amperometric and fluorometric assays.

In the amperometric studies, all the synthetized hybrids showed a completely negligible H_2_S production in the absence of L-Cys, proving that the thiol group acts as a trigger for the release of the sulfide. Betaxolol hybrids (compounds **2a**–**2d**) demonstrated poor H_2_S-releasing properties even in the presence of L-Cys. This behaviour was also confirmed in the fluorometric assay.

Amperometric and fluorometric data showed that molecular hybrids of TBZ (**1a**–**3a**) and ADT-OH (**1b**–**3b**) had a low release of H_2_S compared to HBTA and HPI derivatives (**1c**–**3c** and **1d**–**3d**, respectively). Notably, compounds **1c** (brinzolamide-HBTA), **1d** (brinzolamide-HPI) and **3c** (brimonidine-HPI) were demonstrated to be the best H_2_S- releasing hybrids both in the aqueous solution (in the presence of L-Cys) and in the intracellular environment.

Even if H_2_S reaches low micromolar levels, it is characterized by a hormetic behavior: high concentrations of H_2_S are toxic and H_2_S donors able to donate high amount of H_2_S showed antitumoral activity [27,44]. To obtain benefits from H_2_S donation, the amount of H_2_S should be at low micromolar level, mimicking its physiological production. This low concentration has been demonstrated to activate, for example, Nrf2, to inhibit Nf-kb, and to protect endothelium from harmful stimuli [45,46,47]. Additionally, as reported in the literature [48], the therapeutic concentration range of H_2_S in the ocular tissues is 100 nM-100 μM.

These preliminary results confirm hybridization as a promising strategy in the drug design process. By the synthesis of a new molecular entity through the combination of two or more identical or different drugs, with or without a linker, the aim is to enhance the efficacy of the parent agents [49].

Moreover, based on these results, the idea of combining powerful H_2_S donors such as HBTA or HPI with efficacious IOP-lowering drugs such as prostaglandin analogs could be an interesting, novel perspective to obtain novel antiglaucoma drugs.

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
