# Peer review of "Design, Synthesis and Evaluation of Novel Molecular Hybrids between Antiglaucoma Drugs and H2S Donors"

_ijms, 2022, doi:10.3390/ijms232213804_

Round 1

Reviewer 1 Report

Angela Corvino et al., reporting a manuscript entitled ‘Design, Synthesis and Evaluation of Novel Hydrogen Sulfide Releasing Hybrids as Antiglaucoma Candidates’. In this manuscript they designed and synthetized new molecular hybrids between antiglaucoma drugs and H2S-donors to combine the pharmacological effect of both moieties, providing a synergistic heightened therapy. Brinzolamide, betaxolol and brimonidine were coupled with different H2S donors. The H2S-releasing properties of the new compounds were evaluated in a phosphate buffer solution by amperometric approach and in Human Primary Corneal Epithelial Cells (HCEs) by spectrofluorometric measurements. Experimental data showed that compounds 1c, 1d and 3c were the hybrids with the best properties characterized by a massive and long-lasting production of the gasotransmitter both in the aqueous solution (in the presence of L-Cysteine) and  in the intracellular environments.

However, I have few comments to authors.

1.       Brinzolamide is R-Isomer, Authors used racemic compound. Is there any different if  we use R-isomer instead of racemic. Is there any reason to test racemic compound?

2.       I feel 100 uM concentration is too high to test. Is there any reason for that to test highest concentrations. Would you please comment on cytotoxicity of these compounds at 100 uM.

With these comments I feel this manuscript suitable for publication.

Author Response

  1. Brinzolamide is R-Isomer, Authors used racemic compound. Is there any different if we use R-isomer instead of racemic. Is there any reason to test racemic compound? We thank the reviewer for the suggestion. Actually, we used R-isomer of brinzolamide for the synthesis of H2S donor derivatives, but we didn’t indicate the stereochemistry. Now we have appropriately added this information in the text and specified the stereochemistry also in the scheme 1.
  2. I feel 100 uM concentration is too high to test. Is there any reason for that to test highest concentrations. Would you please comment on cytotoxicity of these compounds at 100 uM? We thank the reviewer for the possibility to discuss the choice of such a concentration: the endpoint of the biological experimental procedure is to demonstrate whether the compounds are able to cross the cell membrane and release H2S inside the cells, monitoring the fluorescence index for a maximum of 60 minutes which represent the incubation time of WSP1 indicated by the datasheet. The test aims at describing the H2S kinetic release from a qualitative point of view during a relative short time of incubation: indeed, we compared the results with a known H2S donor (DADS). Thinking about long lasting incubation (for example 24h) of these compounds, the concentration would be decreased to avoid toxic effect due to the formation of too high amount of H2 This gaseous molecule is able to exert beneficial effect at very low dosage, but the WSP1 test doesn’t allow monitoring for such long periods the fluorescence index.

Reviewer 2 Report

Abstract: The 1st sentence should be re-written because it is not defining glaucoma clearly.

The use of term “synergic” is not correct because the effect of hybrid compound is not greater than the sum of the effect of using only H2S donor or the current antiglaucoma drug.

The use of term “coupled” is not correct for the hybrid compound because H2S donor and the drug is not combined together by a metal ion.

Last sentence in abstract: 1. Incorrect use of “following”. There is a mention of reduction of “adverse effect” which will make sense if authors write about serious side effects with the current antiglaucoma medicines.

Rationale for two different approaches for determining H2S in releasing media and HCE cell lines.

Was the L-cysteine used provided thiols similar to ocular thiol concentration? This is important to include in discussion to show the potential release of H2S in eye.

Line 190: Why did ADT-OH 8 and HPI 10 release H2S in absence of L-cysteine whereas others did not.

Line 209: The best compound is providing Cmax equal to 6.2 micro M. Can this much low concentration be therapeutically relevant or not, should be included in the discussion.

Lines 238-240: It appears self-contradictory. How come more endogenously produced cwill lead to poor fluorescence intensity as WSP-1 is reported by authors to react with H2S.

The use of term “massive release” is not scientifically acceptable; authors may consider term indicating how many fold or % increase in comparison to some control substance.

What is the problems/adverse effects with ACS67?

Author Response

  1. The 1st sentence should be re-written because it is not defining glaucoma clearly.
  2. The use of term “synergic” is not correct because the effect of hybrid compound is not greater than the sum of the effect of using only H2S donor or the current antiglaucoma drug. We thank the reviewer for this observation. Now we have removed the term “synergic”.
  3. The use of term “coupled” is not correct for the hybrid compound because H2S donor and the drug is not combined by a metal ion. We replaced the term “coupled” with “linked to”.
  4. Last sentence in abstract: 1. Incorrect use of “following”. There is a mention of reduction of “adverse effect” which will make sense if authors write about serious side effects with the current antiglaucoma medicines. Thanks for your remark. We modified appropriately the sentence.
  5. Rationale for two different approaches for determining H2S in releasing media and HCE cell lines. The first approach (amperometric) is performed in buffer with or without L-Cysteine 4mM to investigate the mechanism of H2S release. This experiment doesn’t comprehend the use of any biological substrate and has the aim to clarify if the molecule spontaneously donates H2S or has to react with thiols. The WSP1 experiments are completely different: we do not add exogenous thiols but we use cells. Thus, the compounds exploit the thiols that are present inside the cells (we completely remove the medium outside the cells and the experiment is performed in buffer without thiols) and we can predict if the tested compounds are able to cross the cell membrane.
  6. Was the L-cysteine used provided thiols similar to ocular thiol concentration? This is important to include in discussion to show the potential release of H2S in eye. In the amperometric assay (there is no presence of cells) we used an excess of L-cysteine to investigate the mechanism of H2S release. Indeed, many sulfur moieties show thiol dependent H2S donation and for better characterize such behavior in a limited period of incubation (30 minutes) we used L-Cysteine 4mM. The WSP1 experiments are performed using corneal cells without adding exogenous thiols. Thus, the compounds cross the cell membrane and find physiological concentration of thiols that are present in the human corneal tissues. Furthermore, we used non-immortalized corneal cells to be sure that the presence of L-cysteine is the same in normal corneal tissues.
  7. Line 190: Why did ADT-OH 8 and HPI 10 release H2S in absence of L-cysteine whereas others did not. This different behavior is due to the different chemical mechanisms of H2S release. We added the sentence to explain this point. Thank you for your suggestion.
  8. Line 209: The best compound is providing Cmax equal to 6.2 micro M. Can this much low concentration be therapeutically relevant or not, should be included in the discussion. We further discussed this point: H2S is characterized by a hormetic behavior: high concentrations of H2S are toxic and H2S donors able to donate high amount of H2S showed antitumoral activity. To have benefits from H2S donation the amount of H2S should be at low micromolar, mimicking the physiological production. This low concentration has been demonstrated to activate for example Nrf2, to inhibit Nf-kb, and to protect endothelium from harmful stimuli.
  9. Lines 238-240: It appears self-contradictory. How come more endogenously produced cwill lead to poor fluorescence intensity as WSP-1 is reported by authors to react with H2 The WSP1 test allows to measure the H2S that is released inside the cells. If we add only vehicle the fluorescence we monitor reflects the endogenous production of H2S. However, if we add molecules able to donate H2S, the increase of fluorescence index reflects the donation of H2S inside the cells due to the incubation of the molecules. We changed the statement throughout the test so readers can’t get confused.
  10. The use of term “massive release” is not scientifically acceptable; authors may consider term indicating how many fold or % increase in comparison to some control substance. We thank the reviewer for his/her right observation. We changed the statements using a more appropriate language.
  11. What is the problems/adverse effects with ACS67? In the text we cited ACS67 as example of an already developed antiglaucoma drug conjugated to a H2S donor. Starting from the successful results obtained with ACS67, we designed other hybrids that combine H2S donors and other antiglaucoma drugs acting with a different mechanism of action (i.e Adrenergic, carbonic anhydrase inhibitor etc..).

Reviewer 3 Report

1.     The purposes of developing hybrid compounds are to improve the efficacy of antiglaucoma mediations via a synergistic effect, and to reduce side effects. However, in the current manuscript, there is no study or data to support this idea.

2.     According to the Cmax value and fluorescence index, even the three best hybrid compounds do not have a significant enhancement in H2S production when compared to the original H2S-donors HBTA and HPI.

3.     You have indicated that the hybrid compound will undergo disintegration once in the eye or cells, please provide the possible disintegration pathway for your hybrid compounds.

4.     For compound 3a-3d, you only used a succinic spacer, is there any special reasons for not using an acetic spacer?

5.     Compound 1a had an enhanced H2S production compared to the free TBZ moiety. Please provide the possible reasons for that.

6.     Please add the original H2S donors in Figure 3-5 for a side-by-side comparison.

7.     Betaxolol hybrids demonstrated a weak generation of sulfide when compared to brinzolamide and brimonidine derivatives. Please provide the possible reasons from a chemical perspective.

8.     You have indicated that the hybrid compounds exhibit long-lasting release of H2S. Please add the curves of the original H2S donors in Figure 1 for comparison.

9.     Typo: in the abstract, 3c should be 3d; line 64, ‘gasand’ to ‘gas and’; for Cmax values, use a dot instead of a comma; line 213, ‘L-Cyse’ to ‘L-Cys’.

Author Response

  1. The purposes of developing hybrid compounds are to improve the efficacy of antiglaucoma mediations via a synergistic effect, and to reduce side effects. However, in the current manuscript, there is no study or data to support this idea. We thank the reviewer for the observation. The main endpoint of this research has to provide evidence supporting the potential H2S donation inside corneal cells of hybrid molecules developed starting from already used drugs for the treatment of glaucoma and sulfur moieties. To date, the donation of H2S by antiglaucoma H2S donor hybrids using non immortalized corneal cells has never been published and paves the way to further investigate the potential efficacy of such compounds. However, no data about the reduction of side effects have been reported, so we decided to not include this aspect throughout the text. Furthermore, we changed the title in: Design, Synthesis and Evaluation of Novel Hydrogen Sulfide Releasing Hybrids as Potential Antiglaucoma Candidates to highlight the possible efficacy of this molecules.
  2. According to the Cmax value and fluorescence index, even the three best hybrid compounds do not have a significant enhancement in H2S production when compared to the original H2S-donors HBTA and HPI. The purpose of developing H2S donor hybrids is not to obtain a higher increase of H2S donation compared to the original moieties, but to maintain a comparable H2S donation. Indeed, H2S is characterized by a hormetic behavior: high concentrations of H2S are toxic and H2S donors able to donate high amount of H2S showed antitumoral activity. To have benefits from H2S donation the amount of H2S should be at low micromolar, mimicking the physiological production. In this way, we can exploit the effect derived from the donation of H2S and the pharmacological effect due to the drug. This strategy has been widely exploited also for nitric oxide which is probably the most important gaseous mediator in controlling vascular tone. More recently, also H2S donor moieties have been used for synthesizing hybrids for a wide variety of pathologies (Alzheimer disease, osteoporosis, hypertension).
  3. You have indicated that the hybrid compound will undergo disintegration once in the eye or cells, please provide the possible disintegration pathway for your hybrid compounds. These hybrids, as already reported, are suitable for a hydrolytic attack by enzymes, such as esterase and amidase, that are abundant in the biological liquids like tear fluid.
  4. For compound 3a-3d, you only used a succinic spacer, is there any special reasons for not using an acetic spacer? Honestly, we have used different linker to bypass some synthetic problems but the differences in one methylene unit between succinic and acetic linker is completely irrelevant to our goal.
  5. Compound 1a had an enhanced H2S production compared to the free TBZ moiety. Please provide the possible reasons for that. TBZ and the hybrid drug are two different molecules and probably they are able to cross the cell membrane in a different manner. Indeed, the WSP1 test allows to measure the amount of intracellular H2S that is generated after the incubation of the tested compound and it is necessary that the compound has to enter the cell. Furthermore, we also statistically analyzed the data with ONE way ANOVA followed by Bonferroni post test, and there is no significant difference between compound 1a and TBZ.
  6. Please add the original H2S donors in Figure 3-5 for a side-by-side comparison. The release of H2S has been added for a direct comparison between the moiety and the hybrid molecules as requested.
  7. Betaxolol hybrids demonstrated a weak generation of sulfide when compared to brinzolamide and brimonidine derivatives. Please provide the possible reasons from a chemical perspective. This aspect could be correlated to a different solubility and/or cell permeability but it’s also intrinsic in the peculiarity of each different compounds that release H2S. Indeed, CLogP and LogP values, calculated for all the derivatives, show that betaxolol derived hybrids are more hydrophobic and less prone to water solubilization, indispensable condition to the amperometric and intracellular evaluation.
  8. You have indicated that the hybrid compounds exhibit long-lasting release of H2S. Please add the curves of the original H2S donors in Figure 1 for comparison. The amperometric curves of the H2S release by the original moieties have been added.
  9. Typo: in the abstract, 3c should be 3d; line 64, ‘gas and’ to ‘gas and’; for Cmax values, use a dot instead of a comma; line 213, ‘L-Cyse’ to ‘L-Cys’. Typos have been corrected.

Round 2

Reviewer 2 Report

None.

Author Response

We thank the reviewer for approving our MS.

Reviewer 3 Report

1. You indicated that the main endpoint of current research is to prove that the hybrid molecules still have the H2S donation ability inside the cells. If that is the aim of your current study, it's fine and you may not have to provide the data regarding the antiglaucoma activity of your hybrid compounds. However, please make sure that your title, abstract, and introduction clarify your main aim: Molecules resulting from structural hybridization of antiglaucoma drugs and H2S donors still have H2S-releasing abilities. If you would like to emphasize their potential as antiglaucoma agents, it would be better to only mention that in the conclusion as part of your future studies. 

2. Based on your main endpoint of research, the current manuscript should demonstrate the H2S release from hybrids as well as the advantages of hybrid molecules over the combination use of antiglaucoma drugs and H2S donors. The amperometric assay showed a long-lasting release effect of hybrid molecules. However, whether this effect can be transferred to cells is in doubt. The original H2S donors HBTA and HPI  exhibited comparable cumulative H2S release as the candidate hybrids. Please check the H2S release at different time points in the WSP-1 assay to see if the hybrid molecules have long-lasting release ability in cells as well.

3. As you mentioned that the released concentration of H2S should keep at a low micromolar to exert therapeutic effects and avoid toxicity, you also listed two references to support that. However, the two references are both related to the effects of H2S on cancers, and the low micromolar standard is kind of vague. One paper (PMID: 35526614) demonstrates the therapeutic range of H2S in ocular tissues is from 100nM to 200uM. Please check the previous publications or carry out a toxicity assay to provide a more precise therapeutic H2S concentration or concentration range since it is a critical standard to choose candidate hybrids.

Author Response

  1. You indicated that the main endpoint of current research is to prove that the hybrid molecules still have the H2S donation ability inside the cells. If that is the aim of your current study, it's fine and you may not have to provide the data regarding the antiglaucoma activity of your hybrid compounds. However, please make sure that your title, abstract, and introduction clarify your main aim: Molecules resulting from structural hybridization of antiglaucoma drugs and H2S donors still have H2S-releasing abilities. If you would like to emphasize their potential as antiglaucoma agents, it would be better to only mention that in the conclusion as part of your future studies.

We thank the reviewer for the precious suggestion. Consequently, we have changed the title of the manuscript in “Design, Synthesis and Evaluation of Novel Molecular Hybrids between Antiglaucoma drugs and H2S donors” and we have modified the conclusions.

  1. Based on your main endpoint of research, the current manuscript should demonstrate the H2S release from hybrids as well as the advantages of hybrid molecules over the combination use of antiglaucoma drugs and H2S donors. The amperometric assay showed a long-lasting release effect of hybrid molecules. However, whether this effect can be transferred to cells is in doubt. The original H2S donors HBTA and HPI exhibited comparable cumulative H2S release as the candidate hybrids. Please check the H2S release at different time points in the WSP-1 assay to see if the hybrid molecules have long-lasting release ability in cells as well.

We thank the reviewer for his/her observation. The graphs reporting the histograms about the intracellular H2S release after the incubation of the compounds is referred to the area under the curve analysis of the fluorescence increase monitored for 50 minutes. Initially, the graphs of the kinetic were not included throughout the manuscript for not being redundant, however, for a better characterization and for a better comprehension of the results, they will be included in the supplementary data. All the moieties and all the compounds able to release H2S inside the cells, showed a slow increase of the donation of H2S for 50 minutes. Unfortunately, we cannot measure for a longer period the release of H2S as reported in the instruction manual of WSP1.

  1. As you mentioned that the released concentration of H2S should keep at a low micromolar to exert therapeutic effects and avoid toxicity, you also listed two references to support that. However, the two references are both related to the effects of H2S on cancers, and the low micromolar standard is kind of vague. One paper (PMID: 35526614) demonstrates the therapeutic range of H2S in ocular tissues is from 100nM to 200uM. Please check the previous publications or carry out a toxicity assay to provide a more precise therapeutic H2S concentration or concentration range since it is a critical standard to choose candidate hybrids.

The therapeutic range of H2S ranges from 10 to 50 micromolar. This concentration reflects the physiological range of enzymatically produced H2S. However, in pathological conditions the amount of H2S is decreased and exogenous supplementation is needed. Thinking about a therapeutic endpoint, these new hybrid compounds release H2S in a slow manner avoiding high burst release of H2S which could lead to spontaneous pro-inflammatory response and subsequent toxicity. The referenced we included highlight the dual effect of H2S, but refers long lasting treatments (24h or even more). Moreover, as reported in paper kindly suggested, which we added in the text, the therapeutic range of H2S in ocular tissues is from 100nM to 200uM and our compounds tested to 100 uM showed an H2S release in the therapeutic range.

Round 3

Reviewer 3 Report

After reviewing the current manuscript, I would recommend 'Accept in present form.